# Contemporary Pillars of Heart Failure with Reduced Ejection Fraction Medical Therapy

**DOI:** 10.3390/jcm10194409

**Published:** 2021-09-26

**Authors:** Eldad Rahamim, Dean Nachman, Oren Yagel, Merav Yarkoni, Gabby Elbaz-Greener, Offer Amir, Rabea Asleh

**Affiliations:** 1Heart Institute, Hadassah Medical Center, Faculty of Medicine, Hebrew University of Jerusalem, Jerusalem 91120, Israel; Dean@hadassah.org.il (D.N.); oreny@hadassah.org.il (O.Y.); myarkoni@hadassah.org.il (M.Y.); gabbyelbaz@yahoo.com (G.E.-G.); Oamir@hadassah.org.il (O.A.); 2Azrieli Faculty of Medicine, Bar-Ilan University, Safed 1311502, Israel

**Keywords:** heart failure with reduced ejection fraction, novel medical therapy, hospitalization, mortality, morbidity

## Abstract

Heart failure with reduced ejection fraction (HFrEF) is a clinical condition associated with cardiac contractility impairment. HFrEF is a significant public health issue with a high morbidity and mortality burden. Pathological left ventricular (LV) remodeling and progressive dilatation are hallmarks of HFrEF pathogenesis, ultimately leading to adverse clinical outcomes. Therefore, cardiac remodeling attenuation has become a treatment goal and a standard of care over the last three decades. Guideline-directed medical therapy mainly targeting the sympathetic nervous system and the renin–angiotensin–aldosterone system (RAAS) has led to improved survival and a reduction in HF hospitalization in this population. More recently, novel pharmacological therapies targeting other pathways implicated in the pathophysiology of HFrEF have emerged at an exciting rate, with landmark clinical trials demonstrating additive clinical benefits in patients with HFrEF. Among these novel therapies, angiotensin receptor–neprilysin inhibitors (ARNI), sodium–glucose cotransporter-2 inhibitors (SGLT2i), vericiguat (a novel oral guanylate cyclase stimulator), and omecamtiv mecarbil (a selective cardiac myosin activator) have shown improved clinical benefit when added to the traditional standard-of-care medical therapy in HFrEF. These new comprehensive data have led to a remarkable change in the medical therapy paradigm in the setting of HFrEF. This article will review the pivotal studies involving these novel agents and present a suggestive paradigm of pharmacological therapy representing the 2021 European Society of Cardiology (ESC) guidelines for the treatment of chronic HFrEF.

## 1. Introduction

Heart failure (HF) is a significant public health problem affecting millions of individuals globally with high morbidity and mortality rates [1,2]. In people older than 60 years, HF is the leading cardiovascular (CV) reason for hospitalization [3]. HF carries a substantial financial burden in developed countries, with its prevalence continuing to rise over time [4,5,6]. HF has an estimated prevalence of over 37.7 million patients worldwide. In the United States, the estimated prevalence of HF is more than 6 million patients over the age of 20, with over half a million patients newly diagnosed with HF every year [7,8,9,10].

Heart failure is a chronic condition of cardiac functional impairment with various etiologies, pathophysiologies, and clinical presentations. Patients with HF experience a variety of symptoms with a significant impact on their quality of life. Common symptoms include shortness of breath, poor exercise tolerance, and fatigue, which adversely affect daily function [10,11]. HF is differentiated into three major categories based on the left ventricular (LV) ejection fraction (EF). According to the recently released 2021 European Society of Cardiology (ESC) HF guidelines, HF with reduced EF (HFrEF) is defined as HF with EF ≤ 40%. HF with EF between 41% and 49% (previously defined as HF with mid-range HF) is now termed HF with mildly reduced EF (HFmrEF), whereas HF with preserved EF (HFpEF) is defined as HF with EF ≥ 50% (similar to the previous 2016 ESC guidelines) [12,13,14]. 

HFrEF is often accompanied by pathological LV remodeling and dilatation, which leads to adverse outcomes. Reversing cardiac remodeling became a treatment goal and standard of care more than 20 years ago [15]. These patients are recurrently hospitalized, and not rarely, HF patients may require advanced therapies. Guideline-directed medical therapy has led to an increase in survival rates in these patients, with the main target of treatment being the sympathetic nervous system and the renin–angiotensin–aldosterone system (RAAS) [16]. Low effective stroke volume, as seen in HFrEF, typically leads to sympathetic nervous system and RAAS activation. The activation of these systems results in vasoconstriction and fluid retention, thereby contributing to adverse remodeling in HF. 

## 2. Traditional Pillar-Directed Medical Therapy

Sympathetic activation has harmful effects on morbidity and mortality in HF [17]. The beta-adrenergic blocker bisoprolol was found to reduce HF hospitalizations and mortality in the Cardiac Insufficiency Bisoprolol Study (CIBIS) [18]. These benefits have only been found in specific beta-blockers, including metoprolol, carvedilol, and bisoprolol, but not as a class effect. The CIBIS-II trial comparing bisoprolol versus placebo in stable HFrEF patients with New York Heart Association (NYHA) III–IV has demonstrated a 5% absolute risk reduction in all-cause mortality [19]. Similarly, results obtained from HFrEF patients treated with metoprolol versus placebo have shown a 34% relative risk reduction in all-cause mortality per patient-year in the Metoprolol CR/XL Randomized Intervention Trial in Congestive Heart Failure (MERIT-HF), leading to early discontinuation of the trial due to ethical reasons [20]. In 2002, carvedilol was also found to reduce annual mortality rates, HF hospitalizations, and cardiogenic shock compared to placebo in HFrEF patients in the Carvedilol Prospective Randomized Cumulative Survival (COPERNICUS) trial [21]. Head-to-head trials among the three beta-blockers have shown similar benefits without evidence of a preferred agent over the other [22]. 

Angiotensin-converting enzyme (ACE) inhibitors inhibit the conversion of angiotensin I to angiotensin II. The CONSENSUS trial has shown an 18% absolute risk reduction in mortality with enalapril compared with placebo among HFrEF patients with NYHA class IV symptoms after six months of follow-up [23]. Subsequent studies have shown consistent results in patients with NYHA class II and III symptoms [24]. A trend towards increased survival has also been noticed with isosorbide dinitrate and hydralazine. However, in two trials, the V-HeFT and the V-HeFT II, treatment with enalapril has reduced mortality by 7% compared to isosorbide dinitrate and hydralazine [25,26]. The SAVE trial has also shown a 19% decreased mortality with captopril compared with placebo as well as a significant reduction in ventricular dysfunction secondary to ischemia [27]. Based on these studies, ACE inhibitors are considered a class I recommendation in patients with HFrEF [16]. 

Angiotensin receptor blockers (ARBs) inhibit the downstream effects of angiotensin II by blocking its binding to angiotensin I receptors. The Val-HeFT trial has shown that the combination of ACE inhibitors and ARB treatments could cause acute renal failure, and though it resulted in morbidity reduction, no mortality benefit was observed [28]. In 2003, the Candesartan in Heart Failure—Assessment of Reduction in Mortality and Morbidity (CHARM) trial demonstrated a significant reduction in the composite outcome of HF hospitalization or CV mortality in NYHA class II to IV HFrEF patients treated with candesartan as compared to those treated with placebo [29,30]. Accordingly, ARBs are considered as a class I recommendation in symptomatic HFrEF patients [16].

Mineralocorticoid receptor antagonists (MRAs) inhibit the aldosterone receptor, thus hindering sodium and water retention. The Randomized Aldactone Evaluation Study (RALES) has demonstrated an 11% absolute risk reduction in all-cause mortality and a 35% relative risk reduction in HF hospitalization in HFrEF patients when treated with spironolactone versus placebo. This trial has also shown an improved functional capacity in patients with an LVEF < 35% and NYHA class III and IV [31]. Similarly, a 15% relative risk reduction in mortality was found with eplerenone in the Eplerenone Post-Acute Myocardial Infarction Heart Failure Efficacy and Survival Study (EPHESUS) [32]. The Eplerenone in Mild Patients Hospitalization and Survival Study in Heart Failure (EMPHASIS-HF) has shown consistent results, which further emphasized the importance of MRAs in the setting of HFrEF, which is considered as a class I recommendation for patients with EF < 35% and NYHA class II to IV symptoms [33].

The combination of isosorbide dinitrate and hydralazine has shown a trend towards improved survival in the Vasodilator Heart Failure Trial (V-HeFT) compared with prazosin or placebo in HFrEF patients [34]. The recommendation of combining ACE inhibitors with isosorbide dinitrate and hydralazine is based on the African American Heart Failure Trial (A-HeFT), demonstrating a 4% improved survival and a 33% reduction in first HF hospitalization in HFrEF patients [35].

The SHIFT trial (Systolic Heart failure treatment with the If inhibitor ivabradine) involving NYHA II–IV HFrEF patients with a resting heart rate ≥ 70 and at least one HF hospitalization in the previous year demonstrated an 18% relative risk reduction in the composite outcome of HF mortality or hospitalization for HF [36]. The addition of ivabradine is considered a class IIa recommendation by the European Society of Cardiology (ESC) to reduce HF hospitalization in symptomatic patients with NYHA II–III HFrEF [37].

Other pharmacologic treatments, such as digoxin and diuretics, have not been proven to improve survival, although a decrease in HF hospitalization has been observed with digoxin use [38]. Diuretic therapy is currently indicated in patients with volume overload to maintain a volume balance and reduce the risk of rehospitalization. Digoxin can be added to reduce the risk of HF hospitalization in patients who remain symptomatic despite treatments with class I recommendation medical therapies for HFrEF. 

Although HF drugs significantly impact the field, a quarter of patients will still suffer severe symptoms, hospitalizations, and mortality despite optimal treatment. Consequently, novel pharmacological approaches to HF management are vital [7,39,40]. The purpose of this article is to review some of the most recent advancements in pharmacological therapies for heart failure with reduced ejection fraction and future perspectives.

## 3. Novel Pillar-Directed Medical Therapy

A quarter of HF patients will still suffer severe symptoms, hospitalizations, and mortality despite optimal treatment. Consequently, novel pharmacological approaches to HF management are pivotal [9,39,40]. Novel pharmacologic therapies targeting unique pathways involved in the pathogenesis of HFrEF have increasingly become a part of the standard-of-care medical therapy in the past few years. Neprilysin is an endopeptidase responsible for the degradation of natriuretic and other vasoactive peptides under normal conditions. Neprilysin inhibition increases natriuretic peptide levels and other vasodilatory substances and leads to natriuretic and vasodilatory effects. The administration of synthetic natriuretic peptides has not improved outcomes in acute HF [41]. Early trials failed to prove improved outcomes with neprilysin inhibition alone or when combined with ACE inhibitors [42,43]. The combination of neprilysin with ACE inhibitors showed an increased incidence of angioedema, leading to early termination of the trial [44]. The formerly known LCZ696 molecule of sacubitril/valsartan (ARNi) had a unique design of blocking both the renin–angiotensin system and neprilysin activity [45,46,47]. The Prospective Comparison of ARNi with ACE inhibitor to Determine Impact on Global Mortality and Morbidity in Heart Failure (PARADIGM-HF) trial, involving NYHA class II–IV HFrEF patients, was terminated early due to a 20% relative risk reduction in the composite outcome of CV mortality or HF hospitalization, and a 16% relative risk reduction in all-cause mortality with ARNi compared with enalapril in addition to standard therapies in HFrEF [48]. The Comparison of Sacubitril–Valsartan versus Enalapril on Effect on N-terminal (NT) pro B-type-natriuretic peptide (BNP) (NT-proBNP) in Patients Stabilized from an Acute Heart Failure Episode (PIONEER-HF) trial has shown that this treatment is safe and more effective in reducing NT-proBNP levels than ACE inhibitors among patients hospitalized for acute decompensated HF, including ACE inhibitor/ARB-naïve patients [49].

Patients with HFrEF receiving ACE inhibitors or ARBs should be transferred to ARNi when possible, given the greater clinical benefit with ARNi use. ACE inhibitors should be held for 36 h before starting ARNi, while there is no need for this interruption in treatment with ARBs. The initial dosage of ARNi depends on the preceding ACEi/ARB dose, but it is strongly recommended to achieve a maximal dose of 200 mg (sacubitril/valsartan 97/103 mg) twice daily. A lower dose (sacubitril/valsartan 24/26 mg twice daily) should be considered in patients at the age of 75 years or older, with low blood pressure (systolic pressure of 100 to 110 mmHg), estimated GFR < 60 mL/min/1.73 m^2^, or those with significant liver disease. Dosage can be doubled every 2–4 weeks up to the maximally tolerated dose. In an analysis of high-risk patients in the PIONEER-HF study, the reduction in cardiovascular death or rehospitalization after hospitalization for acute decompensated heart failure was similar to the significant risk reduction in the original trial. Compared to enalapril, the initiation of sacubitril/valsartan does not increase adverse events including symptomatic hypotension, worsening renal function, and hyperkalemia [50]. 

### 3.1. Sodium–Glucose Cotransporter-2 Inhibitors (SGLT2i)

Sodium–glucose cotransporter-2 inhibitors (SGLT2i) target the sodium–glucose cotransporter-2 expressed in the early proximal tubules in the kidney, which is responsible for most renal glucose filtration (Figure 1). These medications were initially used as antihyperglycemic agents in patients with type 2 diabetes [51,52]. The mechanisms of the benefits of SGLT2 inhibition are still being elucidated and are likely multifactorial. Besides glycemic control, SGLT2i have additional favorable effects on blood pressure, weight, uric acid concentrations, albuminuria, lipid profile, and hematocrit, as well as direct cardiac effects, including CV and HF benefits [51,53,54,55,56].

Few mechanisms have been proposed to explain the improvement in HF outcomes by SGLT2i. SGLT2i cause osmotic diuresis and natriuresis, which may reduce cardiac preload. Additional suggested mechanisms include direct vasodilation and blood pressure decrease, anti-inflammatory properties, and hematocrit increase [57,58,59]. 

Among patients with type 2 diabetes mellitus (DM), the EMPA-REG trial evaluating empagliflozin in patients with established CVD showed a reduction in HF hospitalization and CV death. Over 10% of the patients participating in this trial had an underlying HF diagnosis, for whom a consistent and significant reduction in HF hospitalization has been observed in the intervention group [60]. The DECLARE TIMI 58 trial [61], involving patients with and without established CVD, showed a lower rate of CV death or hospitalization from HF but not a significant reduction in major CV adverse events (MACE). The CANVAS trial evaluating canagliflozin in patients with type 2 DM and high cardiovascular risk or established CVD (14% with underlying HF) found a lower risk of MACE, hospitalization for HF, and renal protection, but without a significant reduction in mortality [60,62,63].

Given the consistent results showing a remarkable decrease in HF hospitalization in patients with DM and high CV disease risk, the safety and efficacy of SGLT2i in HFrEF patients regardless of having DM were evaluated in the DAPA-HF and EMPEROR-reduced trials. In the DAPA-HF trial [64], dapagliflozin reduced CV mortality by 18% and heart failure hospitalization by 30%, as well as causing a significant reduction in all-cause mortality. In the EMPEROR-reduced trial [65], empagliflozin moderated the risk of the primary composite outcome (CV death and HF hospitalization) by 25%, mostly by a lower risk of HF hospitalization, with no significant decrease in CV death [64,65].

A meta-analysis combining the DAPA-HF and EMPEROR-reduced trials found a 13% reduction in all-cause mortality, a 14% reduction in CV death regardless of background therapy, and a 31% reduction in the risk of the first hospitalization for HF [66]. 

Sotagliflozin is a novel oral anti-diabetic drug with a unique dual-receptor binding affinity for SGLT1 and SGLT2 [67]. The randomized controlled trial (RCT) SOLOIST-WHF evaluated the early use of sotagliflozin in diabetic patients with HFrEF, either before or within three days of discharge after an episode of decompensated HF. The study has demonstrated a significant reduction in the primary combined endpoint of CV death, hospitalization for HF, or urgent visits for HF management. Ertugliflozin is another new SGLT2 inhibitor, which was evaluated for CV outcomes in diabetic patients in the VERTIS-CV trial [68]. Ertugliflozin was non-inferior to placebo for major adverse CV events and reduced hospitalization for HF by 30%. The use of empagliflozin in patients with HF with preserved ejection fraction in the EMPEROR-PRESERVED trial reduced the combined risk of cardiovascular death or hospitalization for heart failure. This benefit was present in patients with and without diabetes. In this trial, a third of patients had EF between 40% and 50%. This benefit is a promising avenue of treatment for patients with HFmrEF [69]. Dapagliflozin is another SGLT2 inhibitor under investigation for patients with HFpEF in the ongoing DELIVER trial.

In the 2021 European Society of Cardiology (ESC) heart failure guidelines, medications that were solely recommended for patients with reduced ejection fraction have shown benefit in HFmrEF and are now indicated as a class IIB recommendation in this cohort. These include ACE inhibitors, ARBs, beta-blockers, MRAs, SGLT2, and sacubitril/valsartan (ARNi). Diuretics are recommended in these patients with congestion to relieve symptoms [14].

### 3.2. Vericiguat

The nitric oxide (NO)–cyclic guanosine monophosphate (cGMP) pathway plays a key role in the regulation of the CV system [70,71]. NO mainly regulates cardiac function by activating soluble guanylate cyclase (sGC), which catalyzes the formation of cGMP, resulting in protein kinase G (PKG) activation [70,72]. PKG modulates cardiac and vascular muscle contraction and energy consumption by decreasing intracellular calcium concentration and myosin light chain activity [73]. The phosphodiesterase (PDE) enzyme family members hydrolyze cGMP, negatively regulating the signaling pathway [74]. The downregulation of the pathway exerts deleterious CV effects, including vascular dysfunction, hypertension, fibrosis, adverse cardiac remodeling, and ultimately the development of HFrEF or HFpEF [70,75,76,77].

Clinical trials of NO donors and PDE inhibitors in HF patients showed no clinical benefit [78,79,80,81,82]. A novel class of medications directly targeting different components of the cGMP pathway, such as sGC agonists, has recently been studied as a therapeutic option for HFrEF. First, sGC activators (e.g., cinaciguat and ataciguat) activate sGC in its oxidized Haem-free form independently of endogenous NO [83,84]. Cinaciguat demonstrated profound hypotension with no clinical, cardiac biomarkers or cardiac index improvement when administered to acute decompensated HF patients in clinical trials, and consequently, its development for HF management was halted [85,86]. It has been speculated that sGC activation irrespective of endogenous NO might be responsible for the substantial debilitating vasodilatory effect of cinaciguat [82].

In contrast to sGC activators, sGC stimulators (e.g., vericiguat and riociguat) potentiate endogenous NO by binding directly to sGC (Figure 2) [84]. In the SOCRATES-Reduced dose defining phase 2b clinical trial, vericiguat was evaluated in 456 participants with worsening HFrEF. At 12 weeks of follow-up, the primary endpoint of decreased NT-proBNP levels was not met, yet a prespecified secondary analysis demonstrated a dose–response relationship. Furthermore, an improvement in the rate of CV death and HF hospitalization as well as a significant improvement in LVEF was noticed in a dose–response manner [87]. The VICTORIA study, a phase 3 RCT, has examined the efficacy and safety of vericiguat in 5050 patients with HF and EF < 45%, elevated NT-proBNP, and recent clinical worsening. Over 10.8 months of follow-up, the primary composite endpoint of CV death or first HF hospitalization was significantly lower in the vericiguat versus the placebo arms (35.5% vs. 38.5%, *p* = 0.02). The difference was mainly driven by a reduction in HF hospitalization, while the difference in the CV death was not significantly different between the two groups. Symptomatic hypotension and syncope rates did not differ between the treatment and control groups (9.1% vs. 7.9%, *p* = 0.12; and 4% vs. 3.5%, *p* = 0.3, respectively) [88]. Following the results of the VICTORIA study, vericiguat received regulatory approval by the FDA for patients with symptomatic chronic HFrEF.

### 3.3. Omecamtiv Mecarbil

Myocardial contraction is a result of chemical energy transformation into mechanical energy. Actin, myosin, and other regulatory proteins generate the force needed for contraction. It also involves ATP hydrolysis and myosin–actin cross bridging, which both play a key role in cardiac contractility [89]. Inotropic drugs increase myocardial contractility through the increase in intracellular cAMP and calcium through different mechanisms. However, the use of inotropes is associated with increased myocardial oxygen consumption and tachyarrhythmia, which may increase mortality as shown in previous studies [89,90,91].

Omecamtiv mecarbil, formerly known as CK-1827452, was the first agent developed to accelerate the transition of the actin–myosin complex from weakly bound to a firmly bound configuration (Figure 3) [90,91,92]. Omecamtiv mecarbil was found to improve cardiac function in patients with HFrEF. However, higher infused doses have led to cardiac ischemia in some cases [92,93]. Safety and tolerability were tested and found not to be different from placebo, including time to angina, exercise duration, or ischemic ECG changes [94]. 

In the ATOMIC-AHF study, patients with reduced ejection fraction admitted with acute decompensated HF were randomized to receive intravenous omecamtiv mecarbil vs. placebo for 48 h. There was no significant effect on dyspnea relief at 6, 24, and 48 h compared with the pooled cohort. There was a benefit on dyspnea relief at 48 h in a supplemental prespecified analysis compared with the paired placebo. There was no effect on 30-day mortality or worsening HF, length of hospitalization, or NT-proBNP levels [95]. 

The COMIC-HF trial studied the pharmacokinetics of modified oral dosing along with safety, tolerability, and echocardiographic and biomarker changes over 20 weeks. This trial showed increased systolic ejection time and stroke volume, reduced heart rate and left ventricular end-systolic volume (LVESV), reduced left ventricular end-diastolic volume (LVEDV), and lower NT-proBNP [96]. The largest phase III trial by Teerlink et al. has shown that HFrEF patients who received omecamtiv mecarbil had a lower incidence of HF events or death from CV causes than patients receiving placebo. CV mortality and all-cause mortality did not differ significantly between the two groups and there was no difference in the Kansas City Cardiomyopathy Questionnaire total symptom score. After 24 weeks, NT-proBNP was 10% lower in the omecamtiv mecarbil group. There was no difference in the frequency of ischemic ventricular arrhythmia events between the groups [97]. A subgroup analysis of the GALACTIC-HF trial by X et al. has shown a more remarkable clinical benefit among HFrEF patients with more pronounced LV systolic dysfunction (represented by lower EF) than those with less severe LV systolic dysfunction [98]. In these trials, there was no increase in angina or ACS events due to omecamtiv mecarbil’s lack of energy demand increase. There was a mild increase in troponin levels of 4 ng/L [97].

These findings create a new avenue of treatments for patients with factors limiting pharmacological therapies, such as low blood pressure and low glomerular filtration rate. This drug can be prescribed to patients with systolic blood pressure as low as 85 mmHg and an estimated glomerular filtration rate of 25 mL/min, creating a new treatment alternative.

The management of HF patients includes four major pillars of pharmacological treatment. The ideal patient will be managed with ARNi, beta-blockers, MRAs, and an SGLT2 inhibitor (Table 1). Beta-blockers and MRAs are longstanding class I recommendations for the treatment of HFrEF. ARNi became a class I recommendation in the 2016 ESC guidelines for the treatment of chronic heart failure for patients with reduced EF [99]. In the 2021 European Society of Cardiology (ESC) HF guidelines, SGLT2 inhibitors became the fourth crucial pillar of treatment for patients with reduced EF (and HFpEF) with or without diabetes mellitus. An emerging new approach will be tailored treatment for HFrEF patients based on their phenotype (Figure 4) [14]. The tailored treatment approach is applied as a general approach. Clinicians can put more emphasis on one of the pillars depending on the clinical scenario. Patients with marked hypertension might benefit from emphasis on treatment with ARNi. Patients with chronic kidney disease (CKD) with hyperkalemia and hypotension might benefit from emphasizing the SGLT2 inhibitors pillar.

## 4. Future Perspectives and Conclusions

Based on a large series of clinical trials, it appears that SGLT2i have the largest effect, hence they have become a class I recommendation in the recent ESC guidelines for chronic HFrEF. Omecamtiv mecarbil and vericiguat have provided less striking results, probably directing their clinical use to specific clinical scenarios, such as omecamtiv mecarbil’s use for patients with lower systolic blood pressure. The future holds tremendous and exciting pharmacologic avenues to be explored. The role of inflammation in morbidity and mortality outcomes in HF patients is a primary area of interest. Genetic analysis of HF patients might shed light on the variation in responses to specific medications, and personalized therapeutic regimens may be suited accordingly. Alongside the tremendous developments described in this manuscript, the rate of patients receiving guideline-directed medical therapy is still low [100]. Therefore, increasing efforts to improve adherence to optimal medical therapy for HFrEF patients is mandatory to achieve maximal clinical benefits.

## Figures and Tables

**Figure 1 jcm-10-04409-f001:**
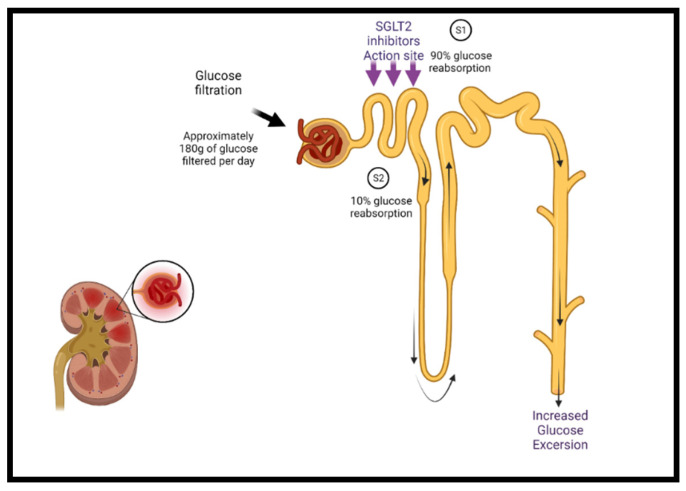
The mechanism of SGLT2 class inhibitors. Under normal conditions, glucose filtration at the nephron reabsorbs glucose back into the bloodstream, approximating 180 g per day. Sodium–glucose cotransporters are present in the proximal convoluted tubule of the nephron (S1 segment) containing SGLT2 and at the distal end (S2 segment) containing SGLT1. SGLT2 is responsible for the reabsorption of 90% of the sodium/glucose, whereas SGLT1 is responsible for the remaining 10%. SGLT2 inhibitors act on SGLT2 proteins at the S1 position and reduce sodium/glucose reabsorption, leading to increased urinary excretion of sodium and glucose, finally lowering blood glucose levels. Additional cardiac-specific mechanisms have been proposed and are currently being studied. SGLT2 = sodium–glucose cotransporter-2.

**Figure 2 jcm-10-04409-f002:**
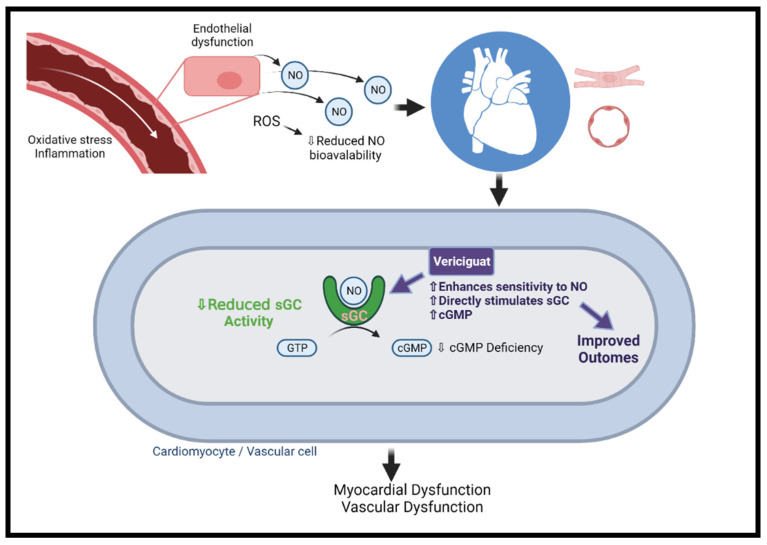
The mechanism of vericiguat. Under normal conditions, NO is generated in endothelial cells and diffuses to neighboring tissues. NO enters vascular/cardiac muscle cells in the heart and stimulates the intracellular receptor sGC to generate cGMP. In HF, there is endothelial dysfunction due to oxidative stress and inflammation and ROS, and ROS reduce NO bioavailability, leading to insufficient activation of sGC. The resulting cGMP deficiency is associated with microvascular dysfunction, cardiomyocyte stiffness, and fibrosis, ultimately leading to myocardial dysfunction. Vericiguat can sensitize sGC and directly stimulate the enzyme to the limited amounts of endogenous NO. cGMP = cyclic guanosine monophosphate; NO = nitric oxide; ROS = reactive oxygen species; sGC = soluble guanylate cyclase. Created with BioRender.com.

**Figure 3 jcm-10-04409-f003:**
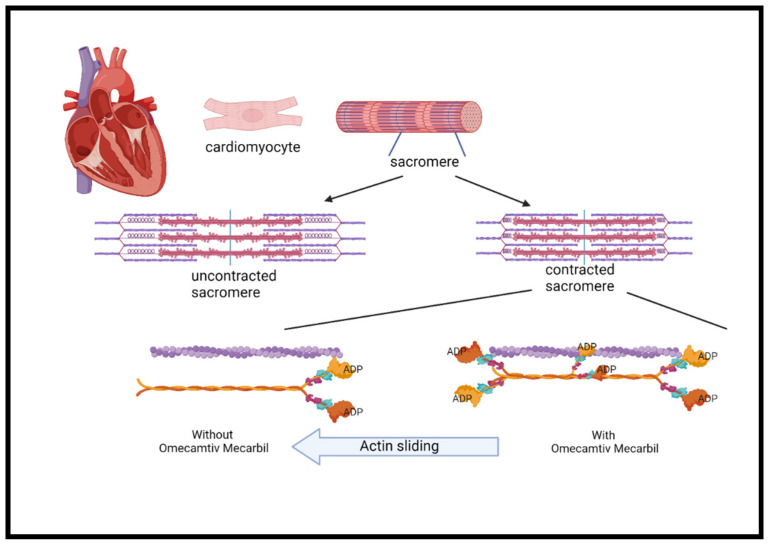
The mechanism of omecamtiv mecarbil. The cardiac myocyte is composed of repeated myofibril units that contain myofilaments. Each unit, termed a sarcomere, is composed of thick and thin filaments, myosin (pink/orange), and actin (purple), respectively. Myosin contains two heads that serve the site of ATPase enzyme that hydrolyzes ATP required for the actin and myosin cross-bridge formation. These heads interact with a binding site on actin and cause the sarcomere length to shorten during contraction. Phosphate is released from ADP to create the force. The more myosin heads containing ADP, the greater the force in each heart contraction. OM binds with highest affinity to myosin heads containing ADP and stabilizes the myosin head in this confirmation 6-fold compared to the other confirmation states. A greater force is produced during each cycle of cardiac contraction. The mechanism has the analogy of hands holding on a rope in which the more hands, the greater the force. ADP = adenosine diphosphate; ATP = adenosine triphosphate; OM = omecamtiv mecarbil.

**Figure 4 jcm-10-04409-f004:**
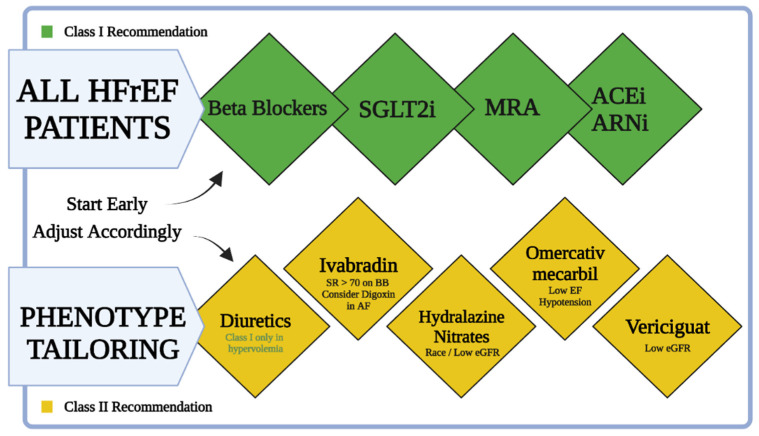
A simplified chart for treatment of HFrEF patients. All HFrEF patients should be started on beta-blockers, SGLT2i, MRAs, and ARNi (or ACEi) as soon as possible after diagnosis. Ivabradine should be considered in patients already on beta-blockers with sinus rhythm (SR) and heart rate > 70. Hydralazine and isosorbide dinitrate should be considered in self-identified black patients with LVEF < 35% despite optimal treatment and may be considered in those who cannot tolerate ACEi, ARB, or ARNi treatment. Omecamtiv mecarbil is a treatment option for patients with hypotension and low eGFR. Vericiguat may be considered in patients with NYHA class II–IV who have worsening HF and low eGFR despite optimal treatment.

**Table 1 jcm-10-04409-t001:** Summary of major HFrEF treatment clinical trials.

Year	Reference	Patient Characteristics	Treatment	Mean Follow Up	Primary Endpoint	Secondary Endpoint	Comments
Beta blockers							
1994	CIBIS [18]	641 patients with chronic HF NYHA III–IV, LVEF < 40%	1.25–5 mg of bisoprolol vs. placebo	1.9 years	Mortality with bisoprolol vs. placebo HR 0.80 (95% CI: 0.56–1.15, *p =* 0.22)	NS in SCD rate, NS mortality rate related to VT/VF, improved functional status of patients on bisoprolol	
1999	CIBIS-II [19]	2647 NYHA III–IV patients, LVEF ≤ 35%, receiving standard therapy with diuretics and ACEi	Bisoprolol 1.25–10 mg vs. placebo	1.3 years	All-cause mortality with bisoprolol vs. placebo HR 0.66 (95% CI: 0.54–0.81, *p* < 0.0001)	Bisoprolol improved sudden deaths HR 0.56 95% (CI: 0.39–0.80, *p =* 0.0011)	Terminated early, after the second interim analysis, because of a significant mortality benefit
1999	MERIT-HF [20]	3991 patients with chronic HF in NYHA functional class II–IV and with LVEF ≤ 40%, stabilized with optimal standard therapy	Metoprolol CR/XL 12.5 mg (NYHA III–IV) or 25.0 mg once daily (NYHA II), target dose 200 mg up-titrated over 8 weeks vs. placebo	1 year	All-cause mortality with metoprolol CR/XL vs. placebo HR 0.66 (95% CI 0.53–0.81, *p =* 0.00009 or adjusted for interim analyses *p =* 0.0062)	Metoprolol CR/XL improved sudden deaths HR 0.59 (CI: 0.45–0.78, *p =* 0.0002) and deaths from worsening HF HR 0.51 (CI: 0.33–0.79, *p =* 0.0023)	Terminated early because of a significant mortality benefit
2002	COPERNICUS [21]	2289 patients with HF symptoms at rest or on minimal exertion and with LVEF < 25%	Carvedilol 3.125 mg twice daily up-titrated to 25 mg twice daily vs. placebo	10.4 months	Combined risk of mortality or CV hospitalization HR 0.73 (95% CI: 0.63–0.86, *p =* 0.00002) Combined risk of mortality or HF hospitalization HR 0.69 (95% CI: 0.59–0.81, *p =* 0.000004)	Carvedilol improved all-cause LOHS HR 0.73 (*p =* 0.0005) and LOHS for HF HR 0.6 (*p* < 0.0001)	
ACE Inhibitors							
1987	CONSENSUS [23]	253 patients with severe CHF NYHA functional class IV	Enalapril initial dose of 5 mg twice daily to a maximal dose of 20 mg twice daily vs. placebo	188 days	Overall 6-month mortality with enalapril vs. placebo HR 0.6 (*p =* 0.002) 1-year mortality with enalapril vs. placebo HR 0.69 (*p =* 0.001) Mortality at the end of the study with enalapril vs. placebo HR 0.73 (*p =* 0.003)	Enalapril improved mortality, reduced heart size, and reduced requirement for other HF medication	Terminated early because of a significant mortality benefit
1992	SAVE [27]	2231 patients with LVEF ≤ 40%, but without overt HF or symptoms of myocardial ischemia	Captopril 12.5 mg up-titrated to 50 mg three times daily vs. placebo	3 years after randomization	All-cause mortality with captopril vs. placebo HR 0.79 (95% CI: 0.65–0.97, *p =* 0.019) CV death HR 0.79 (95% CI: 0.65–0.95, *p =* 0.014) MI HR 0.75 (95% CI: 0.6–0.95, *p =* 0.012)	Captopril reduced risk for the development of severe HF HR 0.63 (95% CI: 0.5–0.8, *p* < 0.001), for CHF requiring hospitalization HR 0.78 (95% CI: 0.63–0.96, *p =* 0.019), and for recurrent MI HR 0.75 (95% CI: 0.6–0.95, *p =* 0.015)	
Angiotensin Receptor Blockers							
2001	Val-HeFT [28]	5010 patients with NYHA class II, III, or IV	Valsartan 40 mg twice daily up-titrated to 160 mg of valsartan or placebo twice daily	23 months	Mortality and morbidity combined endpoint with valsartan vs. placebo HR 0.87 (97.5% CI: 0.77–0.97) Risk with valsartan HR 0.87 (97.5% CI: 0.77–0.97)	Valsartan reduced the risk of HF hospitalization by 27.5% (*p* < 0.001), improved NYHA classification in patients, and relieved worsening outcomes (*p* < 0.001)	Combined endpoints benefit −24% reduction in the rate of adjudicated hospitalizations for worsening HF as a first event in those receiving valsartan
2003	CHARM [30]	4576 CHF patients with NYHA class II–IV with LVEF ≤ 40%	Candesartan 4 mg once daily up-titrated to a maximal dose of 32 mg once daily vs. placebo	Median 40 months	Risk of CV mortality or CHF hospitalization with candesartan vs. placebo HR 0.82 (95% CI: 0.74–0.90) Risk at 1 year, less 30% *p* < 0.001 Risk at 2 years, less 23% *p* < 0.001 All-cause mortality at 1 year, less 33% *p* < 0.001 All-cause mortality at 2 years, less 20% *p =* 0.001	Candesartan improved CHF hospitalization HR 0.76 (95% CI: 0.68–0.85, *p* < 0.001), CV mortality HR 0.84 (95% CI: 0.75–0.95, *p =* 0.005)	
Mineralocorticoid Receptor Antagonists							
1999	RALES [31]	1663 CHF patients in NYHA class III or IV, treated with ACEi and loop diuretic, and had LVEF ≤ 35%	Spironolactone 25 mg once daily up-titrated to 50 mg once daily	2 years	Mortality with spironolactone vs. placebo HR 0.70 (95% CI: 0.60–0.82, *p* < 0.001) by a Cox proportional-hazards model Cardiac mortality HR 0.69 (95% CI, 0.58–0.82, *p* < 0.001)	Spironolactone reduced the risk of cardiac hospitalization HR 0.70 (95% CI: 0.59–0.82, *p* < 0.001), and improved the NYHA classification in patients	The trial was discontinued early
2003	EPHESUS [32]	6200 patients, 3 to 14 days after acute MI with symptomatic HF and LVEF ≤ 40%	Eplerenone 25 mg per day initially, titrated to a maximum of 50 mg per day vs. placebo	16 months	All-cause mortality with eplerenone vs. placebo HR 0.85 (*p =* 0.008), Risk of CV mortality or CV hospitalization HR 0.87 (*p =* 0.002) Risk of all-cause mortality or any hospitalization HR 0.92 (*p =* 0.02)	Eplerenone reduced the risk of SCD HR 0.79 (*p =* 0.03), reduced the risk of HF hospitalization 0.85 (*p =* 0.03), and reduced the episodes of HF hospitalization HR 0.77 (*p =* 0.002)	
2011	EMPHASIS-HF [33]	2737 patients > 55 years, with NYHA class II HF and LVEF ≤ 35%	Eplerenone 25 mg per day initially, titrated to a maximum of 50 mg per day vs. placebo	Median 21 months	All-cause mortality or HF hospitalization with eplerenone vs. placebo HR 0.63 (95% CI: 0.54–0.74, *p* < 0.001)	Eplerenone reduced all-cause mortality or HF hospitalization HR 0.65 (95% CI: 0.55–0.76, *p* < 0.001), reduced all-cause mortality HR 0.76 (95% CI: 0.62–0.93, *p =* 0.008), reduced CV mortality HR 0.76 (95% CI, 0.61–0.94, *p =* 0.01), and reduced HF hospitalization HR 0.58 (95% CI: 0.47–0.70, *p* < 0.001)	The trial was discontinued early
Nitrates and Hydralazine							
1986	V-HeFT [34]	642 chronic CHF patients already taking furosemide and digoxin	40 mg isosorbide dinitrate and 75 mg hydralazine administered four times daily compared to prazosin (5 mg four times daily) and to a placebo	2.3 years	For mortality by two years the risk reduction among patients treated with both hydralazine and isosorbide dinitrate was 34 percent (*p* < 0.028)	The cumulative mortality rates at two years were 25.6 percent in the hydralazine–isosorbide dinitrate group and 34.3 percent in the placebo group; at three years, the mortality rate was 36.2 percent versus 46.9 percent	
2004	A-HeFT [35]	1050 black patients who had NYHA class III or IV HF with dilated ventricles	37.5 mg of hydralazine hydrochloride and 20 mg of isosorbide dinitrate three times daily to a total daily dose of 225 mg of hydralazine hydrochloride and 120 mg of isosorbide dinitrate	10 months	All-cause mortality with combined hydralazine hydrochloride and isosorbide dinitrate vs. placebo HR 0.57 (*p =* 0.01) by the log-rank test	Combined hydralazine hydrochloride and isosorbide dinitrate reduced first HF hospitalizations by 33% (*p =* 0.001) and improved the quality-of-life scores (*p =* 0.02)	Terminated early because of a significant mortality benefit
Angiotensin Receptor Neprilysin Inhibitor							
2014	PARADIGM-HF [48]	8442 patients with class II, III, or IV HF and LVEF ≤ 40%	Treatment with either enalapril (at a dose of 10 mg twice daily) or LCZ696 (at a dose of 200 mg twice daily)	Median 27 months	Risk of CV mortality or HF hospitalization with LCZ696 vs. placebo HR 0.80 (95% CI: 0.73–0.87, *p* < 0.001)	LCZ696 reduced CV mortality HR 0.80 (95% CI: 0.71–0.89, *p* < 0.001), reduced HF hospitalization HR 0.79 (95% CI: 0.71–0.89, *p* < 0.001), and reduced all-cause mortality HR 0.84 (95% CI: 0.76–0.93, *p* < 0.001)	The trial was discontinued early
2019	PIONEER-HF [41]	881 patients with LVEF ≤ 40%, elevated NT-proBNP/BNP, and received a primary diagnosis of acute decompensated HF, including signs and symptoms of fluid overload	The initial dose of sacubitril–valsartan (either 24 mg of sacubitril with 26 mg of valsartan or 49 mg of sacubitril with 51 mg of valsartan as a fixed-dose combination) or enalapril (either 2.5 mg or 5 mg) was administered orally twice daily	8 weeks	Time-averaged reduction in NT-proBNP with sacubitril–valsartan vs. enalapril HR 0.71 (95% CI: 0.63–0.81, *p* < 0.001)	NS worsening renal function, hyperkalemia, and symptomatic hypotension between sacubitril–valsartan vs. enalapril; sacubitril-valsartan reduced the rate of rehospitalization HR 0.56 (CI: 0.37–0.84) and reduced composite of serious clinical events HR 0.54 (CI: 0.37–0.79)	
Hyperpolarization-activated Cyclic Nucleotide (HCN) Channel Inhibitor							
2010	SHIFT [36]	6558 patients with LVEF ≤ 35%, sinus rhythm with heart rate ≥ 70 beats per minute	Ivabradine titrated to a maximum of 75 mg twice daily or matching placebo	Median 22.9 months	Risk of CV mortality or worsening HF hospitalization with ivabradine vs. placebo HR 0.82 (95% CI: 0.75–0.90, *p* < 0.0001) Risk of worsening HF hospitalization HR 0.74 (95% CI: 0.66–0.83, *p* < 0.0001) Risk of HF mortality HR 0.74 (95% CI: 0.58–0.94, *p =* 0.014)	Ivabradine reduced serious adverse events (*p =* 0.025), increased symptomatic bradycardia (*p* < 0.0001), and increased visual side-effects (*p* < 0.0001)	
Sodium–glucose transport-2 (SGLT2) inhibitors							
2015	EMPA-REG [60]	7020 patients with type 2 diabetes with established CV disease	Empagliflozin 10 mg, empagliflozin 25 mg, or placebo (1:1:1)		Primary outcome with empagliflozin vs. placebo HR 0.86 (95.02% CI: 0.74–0.99, *p* < 0.001 for noninferiority, *p =* 0.04 for superiority)	Empagliflozin reduced the key secondary outcome HR 0.89 (95% CI: 0.78–1.01, *p* < 0.001 for noninferiority, *p =* 0.08 for superiority)	
2019	DECLARE TIMI [62]	17,160 patients, including 10,186 without atherosclerotic CV disease but with risk factors	Dapagliflozin 10 mg or matching placebo	Median 4.2 years	Risk of mortality from CV causes or HF hospitalization with dapagliflozin vs. placebo HR 0.83 (95% CI: 0.73–0.95, *p =* 0.005)	Dapagliflozin reduced the incidence of renal composite outcome (>40% decrease in GFR to <60 mL/min/1.73 m^2^, ESRD, or death from renal or CV cause) HR 0.76 (95% CI: 0.67–0.87)	
2017	CANVAS [63]	9734 type 2 diabetes patients and ≥30 years, with a history of symptomatic atherosclerotic CV disease, or ≥50 years with two or more risk factors for CV disease	Canagliflozin 300 mg, 100 mg compared to placebo	188.2 weeks, median 126.1 weeks	Primary outcome with canagliflozin vs. placebo HR 0.86 (95% CI: 0.75–0.97, *p* < 0.001 for noninferiority, *p =* 0.02 for superiority)	Canagliflozin improved the progression of albuminuria HR 0.73 (95% CI: 0.67–0.79) and improved the composite outcome of a sustained 40% reduction in eGFR, the need for renal replacement therapy, or death from renal causes HR 0.60 (95% CI: 0.47–0.77)	
2019	DAPA-HF [64]	4744 patients with LVEF ≤ 40% and NYHA functional class II, III, or IV symptoms	Dapagliflozin 10 mg once daily vs. matching placebo	Median 18.2 months	Risk of mortality from CV causes or HF hospitalization/visit with dapagliflozin vs. placebo HR 0.74 (95% CI: 0.65–0.85, *p* < 0.001)	Dapagliflozin reduced HF hospitalizations or CV mortality HR 0.75 (95% CI: 0.65–0.85, *p* < 0.001)	
2020	EMPEROR-reduced [65]	3730 patients with class II, III, or IV HF and LVEF ≤ 40%	Empagliflozin 10 mg once daily or placebo	Median 16 months	Risk of mortality from CV causes or HF hospitalization with empagliflozin vs. placebo HR 0.75 (95% CI: 0.65–0.86, *p* < 0.001) The effect of empagliflozin was consistent in patients regardless of the presence or absence of diabetes	Empagliflozin reduced HF hospitalizations vs. placebo HR 0.70 (95% CI: 0.58–0.85, *p* < 0.001), slowed the annual decline rate in eGFR (*p* < 0.001), and reduced the risk of serious renal outcomes	
2021	SOLOIST-WHF [67]	1222 type 2 diabetes patients, recently hospitalized due to symptoms of HF, and received treatment with intravenous diuretic therapy	200 mg of sotagliflozin once daily (with a dose increase to 400 mg, depending on side effects) or placebo	Median 9.2 months	Rate of primary endpoint events with sotagliflozin vs. placebo HR 0.67 (95% CI: 0.52–0.85, *p* < 0.001) for an absolute difference of 25.3 events per 100 patient-years (95% CI: 5.1–45.6)	Sotagliflozin reduced CV mortality rates HR 0.84 (95% CI: 0.58–1.22) and reduced all-cause mortality rates, HR 0.82 (95% CI: 0.59–1.14)	
2020	VERTIS-CV [68]	8246 type 2 diabetes patients with atherosclerotic CV disease	5 mg or 15 mg of ertugliflozin or placebo	3.5 years	Risk of mortality from CV causes or HF hospitalization with ertugliflozin vs. placebo HR 0.88 (95.8% CI: 0.75–1.03, *p =* 0.11 for superiority)	Ertugliflozin reduced CV mortality HR 0.92 (95.8% CI: 0.77–1.11) and reduced mortality from renal causes, renal replacement therapy, or doubling of the serum creatinine level HR 0.81 (95.8% CI: 0.63–1.04)	Major adverse CV events occurred in 653 of 5493 ertugliflozin patients (11.9%) vs. 327 of 2745 placebo patients (11.9%) (HR, 0.97; 95.6% CI, 0.85–1.11; *p* < 0.001 for noninferiority)
Soluble guanylate cyclase (sGC) stimulator							
2015	SOCRATES-Reduced [87]	351 clinically stable patients with LVEF < 45% within 4 weeks of a worsening chronic HF event, defined as worsening signs and symptoms of congestion and elevated natriuretic peptide level, requiring hospitalization or outpatient intravenous diuretic	Placebo or 1 of 4 daily target doses of oral vericiguat (1.25 mg, 2.5 mg, 5 mg, 10 mg for 12 weeks)	12 weeks	Δlog-transformed NT-proBNP (baseline to week 12) with pooled vericiguat vs. placebo HR 0.885 (90% CI: 0.73–1.08, *p =* 0.15)	Higher vericiguat doses were associated with greater reductions in NT-proBNP, in a dose–response manner (*p* < 0.02)	Vericiguat 10 mg reduced rates of any adverse event vs. placebo (71.4% vs. 77.2%, respectively)
2020	VICTORIA [88]	5050 patients with chronic HF (New York Heart Association class II, III, or IV) and LVEF < 45%	Vericiguat (target dose, 10 mg once daily) or placebo	Median 10.8 months	Risk of mortality from CV causes or HF hospitalization with vericiguat vs. placebo HR 0.90 (95% CI: 0.82–0.98, *p =* 0.02)	Vericiguat reduced HF hospitalizations HR 0.90 (95% CI: 0.81–1.00) and reduced CV mortality HR 0.93 (95% CI: 0.81–1.06)	
	Cardiac-specific myosin activator						
2016	ATOMIC-AHF [95]	606 patients admitted with acute decompensated HF and LVEF ≤ 40%, dyspnea, and elevated plasma concentrations of natriuretic peptides	Received 48-h intravenous infusion of placebo or omecamtiv mecarbil in 3 sequential, escalating-dose cohorts	30 days	Primary endpoint of dyspnea relief and secondary outcomes with omecamtiv mecarbil (3 dosages) vs. placebo (OM cohort 1, 42%; cohort 2, 47%; cohort 3, 51%; placebo, 41%; *p* = 0.33)	Omecamtiv mecarbil improved dyspnea relief at 48 h (*p =* 0.034) and through 5 days (*p =* 0.038) in the high-dose cohort	NS adverse event profile and tolerability with OM vs. placebo, without increases in ventricular or supraventricular tachyarrhythmias
2016	COSMIC-HF [96]	299 patients with stable, symptomatic chronic HF and LVEF ≤ 40%	Received 25 mg oral omecamtiv mecarbil twice daily (fixed-dose group), 25 mg twice daily titrated to 50 mg twice daily guided by pharmacokinetics (pharmacokinetic titration group), or placebo for 20 weeks	24 weeks	Mean maximum concentration of omecamtiv mecarbil at 12 weeks was 200 ± 71 ng/mL in the fixed-dose group and 318 ± 129 ng/mL in the pharmacokinetic titration group	Omecamtiv mecarbil improved systolic ejection time (95% CI: 18–32, *p* < 0.0001), stroke volume (CI: 0.5–6.7, *p =* 0.0217), LVESD (CI: −2.9 to −0.6, *p =* 0.0027), LVEDD (CI: −2.3 to 0.3, *p =* 0.0128), heart rate (CI: −5.1 to −0.8, *p =* 0.0070), and NT-proBNP concentration in plasma (*p =* 0.0069)	
2021	GALACTIC-HF [97]	8256 patients (inpatients and outpatients) with symptomatic chronic HF and LVEF ≤ 35%	Omecamtiv mecarbil (using pharmacokinetic-guided doses of 25 mg, 37.5 mg, or 50 mg twice daily) or placebo	Median 21.8 months	Risk of CV mortality or HF hospitalization/visit with omecamtiv mecarbil vs. placebo HR 0.92 (95% CI: 0.86–0.99, *p =* 0.03)	NS CV mortality with omecamtiv mecarbil HR 1.01 (95% CI: 0.92–1.11), and NS in the change from baseline on the Kansas City Cardiomyopathy Questionnaire total symptom score	NS frequency of cardiac ischemic and ventricular arrhythmia events with OM vs. placebo

AF—atrial fibrillation, SR—sinus rhythm, ACEi—angiotensin-converting enzyme, ARNi—angiotensin receptor–neprilysin inhibitor, eGFR—estimated glomerular filtration rate, MRA—mineralocorticoid receptor antagonist, SGLT2i—sodium–glucose transport protein inhibitor.

## Data Availability

Not applicable.

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
