# Peer review of "Contemporary Pillars of Heart Failure with Reduced Ejection Fraction Medical Therapy"

_jcm, 2021, doi:10.3390/jcm10194409_

Round 1

Reviewer 1 Report

Thank you for the chance of revising this manuscript. The authors present an elegant overview of the available pharmacological treatment in the field of HFrEF, with a focus on the new drugs that will represent the cornerstone of treatment in the next years of this group of patients.

I have some minor concerns:

  • The authors spoke about Ivabradine in the chapter 3 “Novel Pillars-directed medical therapy”. I would personally add it in the previous paragraph, as the Shift Trial was already published more than 10 years ago and it is already a consolidate treatment.
  • I appreciate the part where the authors speak about individual treatment for each patients with possible drugs that could be added to standard treatment (omecamtiv, vericiguat). I think that a possible tailored treatment might be indicated even for the drugs which are the mainstay of HF treatment. For example preference for SGLT2 inhibitors in patients with marked hypotension or hyperkalemia or viceversa ARNI for patients with concomitant hypertension. I think that the authors might implement the review giving some suggestions for clinicians in this setting.
  • There are many typing mistakes, so I encourage the authors to carefully review the manuscript. Examples: Pag. 2, line 64, remove the , before bisoprololo. Pag 4. Line 161, use as instead of a before anti-hyperglicemic. Pag 5. Line 218, correct Sown with shown. There are also other typing mistakes to be corrected.

Author Response

We appreciate the constructive comments of the Reviewers and have modified the paper accordingly. Below are the point-by-point responses.

Reviewer #1:

Thank you for the chance of revising this manuscript. The authors present an elegant overview of the available pharmacological treatment in the field of HFrEF, with a focus on the new drugs that will represent the cornerstone of treatment in the next years of this group of patients.

RE: We thank the Reviewer for the positive comments.

  1. The authors spoke about Ivabradine in the chapter 3 “Novel Pillars-directed medical therapy”. I would personally add it in the previous paragraph, as the Shift Trial was already published more than 10 years ago and it is already a consolidate treatment.

RE: We appreciate the Reviewer`s comments with which we agree. We have moved this paragraph to the previous paragraph as suggested by the Reviewer.

  1. I appreciate the part where the authors speak about individual treatment for each patients with possible drugs that could be added to standard treatment (omecamtiv, vericiguat). I think that a possible tailored treatment might be indicated even for the drugs which are the mainstay of HF treatment. For example preference for SGLT2 inhibitors in patients with marked hypotension or hyperkalemia or vice versa ARNI for patients with concomitant hypertension. I think that the authors might implement the review giving some suggestions for clinicians in this setting.

    RE: We thank the Reviewer for this excellent comment. We have added the following paragraph in the beginning of the concluding part of the review: “The tailored treatment approach applies as a general approach. Clinicians can put more emphasis on one of the pillars depending on the clinical scenario. Patients with marked hypertension might benefit from emphasis on treatment with ARNi. Patients with chronic kidney disease (CKD) with hyperkalemia and hypotension might benefit from emphasizing the SGLT2 inhibitors pillar.”

  1. There are many typing mistakes, so I encourage the authors to carefully review the manuscript. Examples: Pag. 2, line 64, remove the , before bisoprololo. Pag 4. Line 161, use as instead of a before anti-hyperglicemic. Pag 5. Line 218, correct Sown with shown. There are also other typing mistakes to be corrected.

RE: We appreciate the Reviewer`s comments and apologize for these typing mistakes. These typos have been corrected and the whole manuscript has been reviewed to again to ensure there are no further typos.

Reviewer 2 Report

Minor comment:

Page 8 - the sentence: "The ideal patient will be managed with ARNi, beta blocker, MRA and an SGLT2 329 inhibitor (Table 1)."

It seems to me that It would be better to add the information about an ACEi to ARNi

Author Response

We thank the Reviewer for these insightful comments.

  1. Page 8 - the sentence: "The ideal patient will be managed with ARNi, beta blocker, MRA and an SGLT2 329 inhibitor (Table 1)."
    It seems to me that It would be better to add the information about an ACEi to ARNi

    RE: We thank the Reviewer for this excellent comment. The following paragraph has been included in the revised manuscript considering your excellent suggestion: “Patients with HFrEF receiving ACE inhibitors or ARBs should be transferred to ARNi when possible, given the greater clinical benefit with ARNi use. ACE inhibitors should be held for 36 hours before starting ARNi, while there is no need for this in-terruption in treatment with ARBs. The initial dosage of ARNi depends on the pre-ceding ACEi/ARB dose but it is strongly recommended to achieve a maximal dose of 200 mg (sacubitril/valsartan 97/103 mg) twice daily. A lower dose (sacubitril/valsartan 24/26 mg twice daily) should be considered in patients at age of 75 years or older, with low blood pressure (systolic pressure of 100 to 110 mmHg), estimated GFR <60 ml/min/1.73m2, or those with significant liver disease. Dosage can be doubled every 2-4 weeks up to the maximally tolerated dose. In an analysis of high-risk patients in the PIONEER-HF study, the reduction cardiovascular death or rehospitalization after hospitalization for acute decompensated heart failure was similar to the significant risk reduction in the original trial. Compared to enalapril, the initiation of sacubi-tril/valsartan does not increase adverse events including symptomatic hypotension, worsening renal function and hyperkalemia